# Autoantibodies to Interferons in Infectious Diseases

**DOI:** 10.3390/v15051215

**Published:** 2023-05-22

**Authors:** Eugenia Quiros-Roldan, Alessandra Sottini, Simona Giulia Signorini, Federico Serana, Giorgio Tiecco, Luisa Imberti

**Affiliations:** 1Department of Infectious and Tropical Diseases, ASST Spedali Civili, Brescia and University of Brescia, 25123 Brescia, Italy; eugeniaquiros@yahoo.it (E.Q.-R.); g.tiecco@unibs.it (G.T.); 2Clinical Chemistry Laboratory, ASST Spedali Civili of Brescia, 25123 Brescia, Italy; alessandra.sottini@asst-spedalicivili.it (A.S.); simona.signorini@asst-spedalicivili.it (S.G.S.); federico.serana@asst-spedalicivili.it (F.S.); 3Section of Microbiology, University of Brescia, P. le Spedali Civili, 1, 25123 Brescia, Italy

**Keywords:** antiviral immunity, autoantibodies, COVID-19, cytokines, interferons

## Abstract

Anti-cytokine autoantibodies and, in particular, anti-type I interferons are increasingly described in association with immunodeficient, autoimmune, and immune-dysregulated conditions. Their presence in otherwise healthy individuals may result in a phenotype characterized by a predisposition to infections with several agents. For instance, anti-type I interferon autoantibodies are implicated in Coronavirus Disease 19 (COVID-19) pathogenesis and found preferentially in patients with critical disease. However, autoantibodies were also described in the serum of patients with viral, bacterial, and fungal infections not associated with COVID-19. In this review, we provide an overview of anti-cytokine autoantibodies identified to date and their clinical associations; we also discuss whether they can act as enemies or friends, i.e., are capable of acting in a beneficial or harmful way, and if they may be linked to gender or immunosenescence. Understanding the mechanisms underlying the production of autoantibodies could improve the approach to treating some infections, focusing not only on pathogens, but also on the possibility of a low degree of autoimmunity in patients.

## 1. Introduction

In the 20th century, latent infections in asymptomatic individuals were yet to be well studied and further elucidated. This issue led to the recognition that most infectious agents are lethal for only a small percentage of infected individuals. There is no doubt that an infectious agent is necessary to trigger clinical disease; however, it is just as indisputable that the bacteria, fungi, viruses, and parasites are far from being the only responsible cause of severe disease or death. Therefore, one of the main questions is as follows: what are the characteristics for which a patient suffering from an infectious disease could be life threatening? The enormous interindividual clinical variability observed in most infections appears to be due to genetic and immunological determinants [1]. Apart from genetic background, the great diversity of the human immune system, which is forged based on age, gender-related behaviors, diet, environmental exposure, and microbiome, is a powerful defense against opportunistic pathogens; however, at the same time, it can be the substrate on which immune-associated diseases develop [2].

One of the most important immunological mediators in the natural defense against infectious agents is the group of interferons (IFNs). They are potent cell growth regulators with immunomodulatory activity, but are best known for their antiviral activity [3].

A common phenomenon reported in many globally relevant infections is autoimmunity, with infections and other highly inflammatory diseases being associated with the presence of autoantibodies (aAbs). Therefore, aAbs can be broadly divided into “common” types that are found in apparently healthy individuals and that, through binding a variety of microbial components, provide the first line of defense against infections [4], and “pathogenic” types, which contribute to various immune-mediated diseases.

The large number of studies that previously focused on aAbs targeting cellular antigens, such as dsDNA and lipids, but also immune molecules, such as cytokines, underscores the importance that autoimmunity can play during infections. The role played by aAbs during specific infections is beginning to be an emerging topic of interest, especially after the discovery that anti-type I IFNs (IFN-I) aAbs play a fundamental role in the evolution of severe acute respiratory syndrome coronavirus 2 (SARS-CoV-2) infection [5].

The aim of our paper is to review the functional properties of aAbs produced against cytokines, looking at IFNs in particular, which are important anti-infective components of the immune system, in order to enhance understanding of their role in old and emerging infectious diseases.

## 2. Production of aAbs

### 2.1. Types of aAbs

The production of antibodies (Abs) is a vital way in which the adaptive immune system works either to recognize and neutralize or eliminate antigens and pathogens. Although thought to be absent in healthy individuals, due to the immune tolerance mechanism [6], the common Abs that react with self-molecules are found in healthy subjects [7]. They also fulfill the definition of aAb since they are self-reactive, rather than self-specific, and characterized by a broad reactivity directed against well-conserved public epitopes [8].

A type of common aAbs can be generated when molecules of infectious agents share similarity with foreign and self-peptides. Through a mechanism defined as “molecular mimicry” [9], these proteins may activate self-reactive T or B cells, thus producing cross-reactive Abs. For instance, the Abs produced against a viral phosphoprotein of measles or type 1 Herpes simplex viruses (HSV) cross-react with an intermediate filament protein of human cells [10]. Similarly, the significant sequence homology between Coxsackie virus P2-C protein and glutamate decarboxylase in humans may trigger type 1 diabetes [11], while ankylosing spondylitis, systemic lupus erythematosus (SLE), and Lyme disease may be induced by antigens such as pulD from *Klebsiella* sp., OSP-A from *Borrelia* sp., and nuclear antigen-1 from Epstein–Barr virus [12,13]. More recently, others potential molecular mimicry candidates were identified using bioinformatics techniques [14,15].

Another class of aAbs includes those defined as “natural” because they are present in the blood without any evident antigenic stimulation. Accordingly, this type of aAbs was even identified in mice raised under germ-free conditions [16]. Unlike adaptive Abs, these aAbs are synthesized by CD20+CD27+CD43+CD70−B1 lymphocytes and marginal-zone B cells [17,18] and do not undergo affinity maturation via antigen stimulation or extensive somatic mutation [19]. Their functions are not clear, though they may play a role in the maintenance of immune homeostasis, regulation of the immune response, resistance to infections, and transport and functional modulation of biologically active molecules [19,20].

The presence of aAbs rarely induces improvement in the disease, except in the cases of anti-cytokine aAbs associated with mild autoimmune diseases or specific anti-cancer aAbs that can be found in cancer patients with better survival [21,22,23,24,25]. On the contrary, they often cause adverse effects, playing a major role in several infectious, autoimmune, cardiovascular, neurological, and neurodegenerative diseases, as well as in metabolic dysfunction and cancers [26,27,28,29,30,31,32,33].

Anti-cytokine aAbs were initially described in an increasing number of primary immunodeficiencies with autoimmune features, especially autoimmune polyendocrine syndrome type I (APS-1), which is a disease of defective T cell-mediated central tolerance. However, these aAbs were proposed as an emerging alternative pathological mechanism leading to impaired immune response and susceptibility to infections.

### 2.2. Anti-Cytokine aAbs

In recent years, life-threatening diseases caused by anti-cytokine aAbs received widespread attention. Anti-cytokine aAbs found in healthy individuals [34,35] often are not strongly inhibitory, not necessarily associated with a respective neutralizing activity, and typically detected at lower binding titers [36,37,38,39]. Therefore, they seem to be a physiologic mechanism used to control the immune response [22]. In contrast, pathogenic anti-cytokine aAbs, which are usually polyclonal IgGs, may affect cytokine biology through diminishing or augmenting signaling or altering their half-lives in the circulation [22,38,40,41,42,43]. Anti-cytokines aAbs were found in patients with SLE, Sjogren’s syndrome, and rheumatoid arthritis [43]. In addition, diseases due to aAbs targeting specific cytokines or cytokine pathways, which are classified in a unique category termed “phenocopies of primary immunodeficiency”, comprise acquired immunodeficiency characterized by the presence of some anti-cytokine aAbs, notably to IFN-γ, interleukin (IL)-6, IL-17, IL-22, and granulocyte macrophage colony stimulating factor (GM-CSF). These diseases were found in particular, but not exclusively, in adult patients who showed phenotypic manifestations similar to those that occur due to pathogenic variants in genes encoding for the specific cytokines, their receptors, or molecules mediating cytokine signal transduction [44]. In addition, aAbs against pro-inflammatory cytokines are also found in multiple sclerosis that affect young adults [22]; however, their biological role is not yet clarified.

Anti-cytokine aAbs received great attention in recent years, especially in explaining the enormous phenotypic variability in infections, as well as the different incidence and inter-individual response variability. The development of several infectious diseases are described to be associated with the presence of aAbs targeting a number of cytokines [45]. Anti-IL-2 aAbs were found in patients with human immunodeficiency virus (HIV) infection [46]. Anti-GM-CSF aAbs were detected in patients with cryptococcosis, nocardiosis, non-tuberculous mycobacteria (NTM), and histoplasmosis [47,48], while anti-IL-6 aAbs were associated with severe bacterial infections, such as *Escherichia coli*, *Streptococcus intermedius*, and *Staphylococcus aureus* [49], including staphylococcal sepsis [50]. IL-17A, IL-17F, and IL-22 are considered important in mucosal immunity, principally in chronic mucocutaneous candidiasis [51].

Therefore, the presence of anti-cytokine aAbs can have severe consequences and cause highly varied manifestations.

Very recent anti-cytokine aAbs were found in >50% of critically ill patients with non-SARS-CoV-2 infections, i.e., caused by other viral and fungal pathogens, as well as known or suspected bacterial pathogens [52]. These aAbs were far more common in infected versus uninfected patients and, importantly, were seen not only in multiple respiratory viral infections, but also in the non-respiratory bacterial infections observed in patients admitted to the intensive care unit (ICU). Moreover, while most of these aAbs were present at the onset of infections, some can emerge over time and persist for at least 28 days after infection.

## 3. Role of IFNs and Anti-IFN aAbs in Infectious Diseases

### 3.1. IFNs in Infectious Diseases

There are three families of IFNs (Table 1): IFN-I (mainly IFN-α, IFN-β, and IFN-ω), IFN type II (IFN-II; IFN-γ), and IFN type III (IFN-III; IFN-λ).

In humans, they constitute the first line of defense in response to invading pathogens. Indeed, IFN-I signal transduction pathways were previously identified as a critical factor limiting cytomegalovirus (CMV) infection and replication [55]. IFN-I contributes to the control of latent HSV infections, especially those caused by Alphaherpesvirinae HSV and varicella zoster virus (VZV). Similarly, it is crucial for HIV-1 infection of monocytes and macrophages [56,57], and its deficiency is related to significant impairment of the immune response during productive HIV-1 infection and infection latency [58]. Finally, dysregulation of the IFN-I signaling pathway by *Mycobacterium tuberculosis* (MT) also leads to exacerbation of HIV-1 infection in macrophages [59]. These data underline the importance of IFN-I responses towards HIV-1 infection [60] and, probably, in co-infections of HIV-1-infected patients with CMV and MT. However, the effects of IFN-I on the outcome of different infections are very complex, with both protective and detriment effects according to type of micro-organism and types of IFNs involved [61]. One of most representative examples is MT infection, for which IFN-I signaling demonstrated both pathogenic and protective roles [62].

IFN-II, as a key player in driving cellular immunity, can orchestrate numerous protective functions to heighten immune responses during several infectious diseases. Indeed, it has important immunomodulatory effects because it increases antigen processing and presentation, enables leukocyte trafficking, induces an anti-viral state, boosts the antimicrobial functions, and affects cellular proliferation and apoptosis [63]. IFN-II is important in endowing protection against bacterial infections, such as *Chlamydia* [64,65,66], *Staphylococcus aureus* [67], MT [68], NTM [69], *Salmonella* [70], and *Listeria* [71]. In addition, the protective benefits of IFN-II can be observed in the context of viral infections because its production via Natural Killer (NK) cells can successfully limit Hepatitis C virus proliferation in HIV-1-infected patients [72], while IFN-II treatment enhances survival of neurons infected with VZV [73]. Finally, IFN-II plays a pivotal role in host resistance to parasite invasions, which happen during *Leishmania* and *Toxoplasma* infections [63].

The most recently found member of IFNs, IFN-III was originally thought to act in parallel to IFN-I to activate compartmentalized antiviral responses [74]. Subsequent studies provided increasing evidence for distinct roles for each IFN family [75], and it seems that IFN-III can be also a critical instructor of antifungal neutrophil responses [76] and main players in protecting intestinal cells against enteric virus infections [77].

### 3.2. aAbs against IFNs in Non-Infectious Diseases

During the initial stages of an infection, there is a balance between mechanisms that promote or inhibit micro-organism invasion. Usually, the inhibiting mechanisms are able to clear the infection; however, sometimes, some bacteria or viruses can emerge and evade the host interferon response. The IFN response that helps host cells fight off invading pathogens occurs in two phases: an initial intracellular phase, in which infected cells produce IFNs, and an intercellular phase, in which infection-induced IFNs are secreted into the extracellular environment. Secreted IFNs bind to IFN receptors on surrounding cells, leading to the synthesis of proteins and more IFNs and resulting in a rapid clearance of pathogens [78]. When chronic, recurrent, hard-to-control, or unusually serious infections of common pathogens compared with the normal population occur, a potential deficiency in the IFN defense can be considered. Regardless of the type of IFN, the production of high titers of anti-IFN aAbs in serum interrupts the activation of the downstream response pathway through blocking the combination between IFNs and their receptor, resulting in increased infection rates [45]. Therefore, anti-IFN aAbs occur in previously healthy people who develop chronic, recurring, and difficult-to-control infections [45]. Although aAbs neutralizing the activity of IFN-I were detected for the first time about four decades ago in a 77-year-old woman with disseminated Herpes zoster virus [79], their presence was considered clinically silent in the general population until the Coronavirus Disease 19 (COVID-19) pandemic caused by SARS-CoV-2.

There are multiple methods for detecting aAbs against, including preferentially neutralizing assay [80], radioimmunoprecipitation and real time PCR [81], time-resolved immunofluorometric assay [82], radioimmunoassay [83], magnetic-beads-based assay [84], cell-based autoantibody assay (CBAA) [85], microarray-based assay [86], and homemade [5,87] and commercially available enzyme-linked immunosorbent assays (ELISA). These methods were developed and improved in recent years and are used principally to detect anti-IFN-I aAbs for diagnostic purposes.

Anti-IFN aAbs were initially found in patients treated with IFN-α or IFN-β [88], before being found in patients with chronic graft-versus-host disease following allogeneic bone marrow transplantation [89], myasthenia gravis [90], or thymoma [91] as well as in some women with SLE [43]. They were also previously detected in most patients with APS-1 [92]; in some patients with combined immunodeficiency due to hypomorphic recombination activating genes (RAG)-1 or RAG-2, Omenn’s syndrome, leaky severe combined immunodeficiency, T-cell lymphopenia, and ataxia-telangiectasia [93]; in men with X-linked enteropathic polyendocrine immunodysregulation and forkhead box P3 (FOXP3) mutations [94]; and in women with incontinentia pigmenti and heterozygous “null” mutations in X-linked NEMO syndrome [5].

The biological mechanism behind anti-IFN-II aAbs’ role in diseases remains unclear. Several studies showed that they are able to block the binding of IFN-γ to its receptor, which inhibits the early signal transduction, as well as the downstream biological consequences of IFN-II binding, which include the upregulation of tumor necrosis factor (TNF)-α and IL-12 production [95]. Accordingly, neutralizing anti-IFN-II aAbs are the cause of adult-onset immunodeficiency (AOID) and the associated Sweet syndrome [96], which are both characterized by the increased risk of opportunistic infections (OIs), as reported in the next chapter. The anti-IFN-II aAbs present in some healthy subjects did not show neutralizing activity or effects on IL-12 production [97].

To our knowledge, anti-IFN-III aAbs have so far only been studied in patients with rheumatologic diseases; however, in these patients, they did not exhibit neutralizing activity [43].

### 3.3. aAbs against IFNs in Infectious Diseases

IFNs have immunoregulatory functions during infection and immune responses, and, accordingly, defective activity of IFNs significantly contributes to infections’ severity [1]. In particular, since IFN-I play a role in tightening barriers at mucosal interfaces, Abs that neutralize its activity can be linked with a negative outcome during respiratory infections [98].

It should be emphasized that aAbs directed against human IFN-α were first observed in a patient with VZV in 1981 [99]; only 20 years later, they were described in relation to the emergence of anti-IFN-I aAbs in HCV-infected patients during IFN-α treatment. Moreover, their presence was predictive of breakthrough despite an initial response to treatment, suggesting the pathogenic role of these aAbs in the loss efficacy of IFN-α therapy and HCV reactivation [100,101]. After a further 20 years, scientific interest moved to the role of anti-IFN aAbs in the course of respiratory infections. Indeed, recent data reported that neutralizing anti-IFN-I aAbs (against IFN-α2 alone or with IFN-ω) can be involved in critical influenza pneumonia, as they are present in about 5% of cases of life-threatening respiratory infections. After adjustments for age and sex, the presence of high concentrations of both IFN-α2- and IFN-ω-neutralizing aAbs were found to induce the highest risk of critical influenza pneumonia in patients <70 years old [102]. Similarly, these aAbs seem to play a role in complicating the severe respiratory syndrome, with a high fatality rate caused by Middle East respiratory coronavirus (MERS CoV) recorded. The 93.3% of MERS CoV-infected patients with anti-IFN-I aAbs (IFN-α2, IFN-β, and/or IFN-ω) were critically ill and needed to be admitted to the ICU, compared to just 66% of patients without aAbs. However, the presence of anti-IFN-I aAbs was not associated with different clinical outcomes or responses to treatment with IFN-β1b or antiviral drugs [103].

On the contrary, a low incidence (1.1%) of anti-IFN-I aAbs was found in a cohort of critically ill patients with acute respiratory failure. The study, however, included patients with both infectious (rhinovirus, influenza, parainfluenza, and seasonal coronavirus infections) and non-infectious etiologies [104]. The importance of anti-IFN-I aAbs during infections with severe pulmonary involvement was confirmed based on the fact that no increased prevalence of neutralizing anti-IFN-I aAbs were found in patients with idiopathic pulmonary fibrosis compared with general population [105].

IFN-I seems have a role during Flaviviruses infections, including those caused by yellow fever virus (YFV), dengue, West Nile, and Zika viruses [98], though no studies were performed that searched for anti-IFN-I aAbs during these emerging infections. However, anti-IFN-I aAbs were detected in one third of subjects with severe adverse events following vaccination for YFV, albeit only for vaccinations that used an attenuated alive virus. High titers of circulating aAbs against at least 14 of the several IFN-I were found. The authors demonstrated their IFN-I-neutralizing activity in vitro, blocking the protective effect of IFN-α2 against YFV vaccine strains [106]. Of interest is the potential link between the presence of anti-IFN-I aAbs and possible reactivation of latent virus infections, particularly HSV (CMV, HSV-1/2, or both) in critically ill COVID-19 patients [107].

Finally, our group found an elevated prevalence (11.6%) of anti-IFN-I aAbs in HIV-1-infected patients with OIs (neutralizing activity was not determined); however, no statistically significant differences were found for viro/immunological characteristics (CD4 and CD8 cell counts and viral load) between patients with and without anti-IFN-I aAbs (submitted). While the anti-IFN-II aAbs were first identified in a group of HIV-infected patients [40], those with neutralizing activity were first described in 2004 in the context of selective susceptibility to NTM infection [108]. Accordingly, neutralizing anti-IFN-II aAbs were detected in 88% of patients with multiple OIs in Asia [109] and in 62% of subjects with disseminated NTM and without HIV infection [110]. Nevertheless, the association between their presence and intracellular infections, especially in the contest of the onset of immunodeficiency, was clearly established years later. Indeed, it is well known that anti-IFN-II aAbs play a critical role in the pathogenesis of AOID, which is also referred to as AIDS-like syndrome [111], in which infectious diseases caused by opportunistic pathogens were firstly reported in adults without known immunodeficiency. Although genetic factors (HLA-DQB1*05:01 andHLA-DQB1*05:02) are associated with a very high risk of critical AOID and environmental exposure contributes to AOID, the autoimmunity caused by anti-IFN-II aAbs is critical for the syndrome’s pathogenesis. It is also known that anti-IFN-II aAbs titers were strongly associated with the severity of infections, which was related to their neutralizing activity [112].

Recent findings confirmed the presence of IFN-II aAbs in the course of disseminated NTM disease, non-typhoid *Salmonella*, *Cryptococcus*, and VZV infections, particularly in Asian populations with AOID [113]. Interestingly, NTM is the most common pathogenic micro-organism in patients with AOID showing high titers of anti-IFN-II aAbs [113]. High titers of highly neutralizing anti-IFN-II aAbs were also reported by several groups in sporadic cases of disease caused by low-virulence *Mycobacteria* and MT, as well as after Bacillus Calmette–Guérin vaccination [44]. An association between anti-IFN-II aAbs and infections caused by other opportunistic pathogens, including those caused by other bacteria, e.g., *Burkholderia,* fungi, e.g., *Penicillium*, *Histoplasma*, and *Candida* sp., and viruses, in particular VZV and CMV, were also described in other studies [114,115].

To our knowledge, the presence of anti-IFN-III aAbs were described in patients with non-SARS-CoV-2 respiratory infections, where they seemed to display a neutralizing activity [52].

### 3.4. aAbs against IFN-I in COVID-19

The COVID-19 pandemic began at the end of January 2020, causing almost 7 million deaths worldwide and more than 680 million infections [116]. SARS-CoV-2 can cause infections of very different severities, ranging from asymptomatic forms to extremely serious cases that require hospitalization in ICU and cause death [117]. As IFNs represent the first line of defense during the early phase of viral infection, the levels of these cytokines were described as relevant in determining the outcome of SARS-CoV-2 infection. In particular, low levels of IFNs in the lungs or peripheral blood renders SARS-CoV-2 capable of evading innate recognition [118,119,120]. IFN-I levels may change due to many factors, i.e., age [121], gender [122], genetic defects in IFN-related encoding genes [123], and the presence of neutralizing anti-IFN-I aAbs [5]. Up to 10% of elderly patients were positive for anti-IFN-I aAbs (neutralizing IFN-α2, β and/or ω); this percentage may increase with COVID-19 severity [5,124,125]. The presence of anti-IFN-I aAbs mainly correlates to severity of COVID-19 in males over 65 years old [87,126]. Many compelling studies confirmed the link between anti-IFN-I aAbs and COVID-19 severity (Table 2).

In a recent meta-analysis including more than 7700 patients, the positive rate of anti-IFN-I aAbs was found to be 5% (95% CI, 3–8%); however, this rate reached 10% (95% CI, 7–14%) when analysis was restricted to patients with severe infections [150].

The evidence that anti-IFN-I aAbs are capable of altering the course of COVID-19 through perturbing the immune response to SARS-CoV-2 and tissue homeostasis was also provided through data obtained regarding these aAbs with mouse surrogates, which led to increased disease severity in a mouse model of SARS-CoV-2 infection [128].

Nonetheless, different data are provided by a recent investigation that demonstrated the presence of autoreactive polyclonal B-cell activation and aAbs production, but did not demonstrate a correlation between anti-IFN-I aAbs levels and COVID-19 severity [151]. These conflicting results could be explained via different assays used to detect aAbs with high or low affinity, different patient populations tested, or the different contribution to disease outcomes of pre-existing or infection-induced neutralizing anti-IFN-I aAbs.

We also studied whether the presence of anti-IFN-I aAbs could have a role in SARS-CoV-2 breakthrough infections in vaccinated patients. Breakthroughs were reported worldwide, with most of them being asymptomatic or mild cases; the few breakthrough critical cases were mainly described in immune-depressed patients. A total of 20% of these breakthroughs can occur in patients with normal antibody response to the vaccine who also carry aAbs neutralizing IFN-α2 and/or IFN-ω [87].

Finally, the presence of anti-IFN-I aAbs was previously proposed as a possible driver of post-acute COVID-19, which is also known as “long COVID” [152,153,154], in which a persistent immune response seems be the inducing mechanism [155]. However, anti- IFN-α2 aAbs were uncommon in long-COVID patients [156].

Until now, the relationship between anti-IFN-II aAbs and COVID-19 severity was little explored. Only a pilot study in Taiwan described the presence of anti-IFN-II aAbs with neutralizing activity in the 18% of COVID-19 patients with severe/critical illness. The prevalence was statistically higher compared with non-severe COVID-19 patients or healthy controls. Moreover, median titers of anti-IFN-II aAbs were higher in severe/critical patients than in patients with mild/moderate disease or healthy controls [157].

Anti-IFN-III aAbs were recently detected via the Molecular Indexing of Proteins by Self-Assembly technology in patients with life-threatening COVID-19, though they were not detected in plasma samples of healthy subjects or convalescent plasma from non-hospitalized individuals with COVID-19 [158].

## 4. Significance of the Production of aAbs against IFNs

### 4.1. Is There a Gender Bias for aAbs Production?

Recent results demonstrated that aAbs production among healthy subjects did not show a gender bias because the median numbers and the weighted prevalence of 77 common aAbs were similar between males and females [159]. This result stands in contrast to the evidences that autoimmune diseases disproportionally affect females compared with males. While the risk of contracting autoimmune diseases is up to four times greater in women than in men, the mechanism for this sex bias is still obscure. Several hypotheses were previously proposed, including that women have an evolutionarily conserved tendency toward an enhanced activation of B cells resulting in higher levels of antibody production, which may be responsible for the increased incidence of antibody-driven autoimmune diseases [160]. This suggestion agreed with the fact that autoimmune diseases in females are associated with antibody-mediated pathology, whereas in males they are preferentially associated with acute inflammation [161].

Gender differences in circulating anti-cytokines aAbs were observed in atherosclerosis, with an upregulation of anti-TNF-α, anti-IL-1α, and anti-IL-1β IgG levels are more likely to occur in female than in male patients [162]. Similarly, among patients with acquired pulmonary alveolar proteinosis, which is an ultrarare autoimmune disease characterized by accumulation of excess surfactant in the alveoli, leading to pulmonary insufficiency, men are predominantly affected (male:female ratio of 65:1), and high levels of aAbs that neutralize GM-CSF signaling were detected [163]. In addition, 10 of 14 patients with severe disseminated NTB and no other evidence of immunodeficiency who produced anti-IFN-II aAbs were females [164]. Finally, only 2.6% of females with life-threatening COVID-19 showed anti-IFN-I aAbs, compared to 12.5% of males [5]. In particular, neutralizing anti-IFN-I aAbs were detected in 94% of males with critical COVID-19 pneumonia. The recently published meta-analysis [150] confirmed that the generally (not calculated according to disease severity) higher prevalence of neutralizing aAbs in males (5%) than in females (2%). Different data were obtained regarding 130 critically ill COVID-19 Swiss patients, in which 11.3% of males and 13.0% of females showed detectable anti-IFN-α2 aAbs, while 7.5% of males and 8.7% of females presented anti-IFN-ω aAbs in their plasma, which were not present in plasma of 130 healthy donors [107]. Moreover, in our laboratory, we tested the presence of anti-IFN-I aAbs in 349 critically ill male and female COVID-19 patients (Table 3; previously unpublished data). We found a percentage of anti-IFN-I aAbs-positive patients comparable to that of previously published studies (see Table 2); however, unlike Busnadiego et al. [107], we did not observe a gender difference in our patients. This result can be due to the fact that females may carry more non-neutralizing anti-IFN-I aAbs and that the female age was slightly higher than that of males. Furthermore, females also appeared to have a more critical disease, since 50% (vs. 35% of males) of them died, although this difference was not significant.

Notably, anti-IL-6 aAbs were predominantly elevated in asymptomatic COVID-19 females [165]. However, the reported data, together with those included in another study [166], revealed that despite classic autoimmune diseases being more prevalent in females, a paradoxical male predominance of autoimmune activation illness is present in the setting of severe COVID-19.

### 4.2. Are Infection-Related aAbs Associated to Immunosenescence?

Age-associated changes in the immune system heavily increase the risk of bacterial and viral infections in the elderly [167]; however, the relationship between aAbs production, aging, and infectious diseases has not yet been formally demonstrated. Moreover, reasons for the generation of aAbs are not exactly clarified, although several hypotheses were proposed. Potential hypotheses are as follows: (a) tolerance defects and inflammation; (b) modification of antigen expression; (c) changes in exposure or presentation of antigens; (d) cellular death mechanisms; (e) combination of genetic and environmental factors (e.g., simultaneous exposure to microorganisms with certain toxins and hazardous chemicals); and (f) infections with viral proteins with sequences similar to a human protein (in around 20 autoimmune diseases, aAbs are generated due to cross-reactivity to infectious agent proteins) [168,169]. The thymic involution, which is a naturally occurring part of the aging process that leads to reduced thymus activity, also increased the likelihood of high autoimmune incidence [170]. Interestingly, the rate of thymic involution can be regulated via numerous growth hormones and sex steroids, as well as via metabolic activity, and involution appears to occur more rapidly in males than females [170], thus further supporting a gender bias in aAbs induction (see previous section). Therefore, those factors that reduce thymus activity, including some cancers, age, sex or certain other diseases/disorders and lifestyles, which are associated with immunosenescence, could also increase individual risk of developing aAbs.

In the last few years, increased levels of aAbs were also associated with the accumulation of Age-Associated B Cells (ABCs) [171], which are one of the immune changes that characterize the immunosenescence [172]. Indeed, ABCs were observed to secrete aAbs [173], which were first described more than 50 years ago [174,175]. It is also of note that ABC-like cells producing aAbs were identified during HCV infection [176], and that ABCs expand in the presence of viral infections not associated with autoimmune disease, such as murine CMV [177], influenza virus [178], HIV-1 [179], and SARS-CoV-2 [180]. In this last infection, the expansion of ABCs may be responsible for the increased levels of anti-IFN-I aAbs, which are associated with the higher risk of critical COVID-19 infection in the elderly population [5,87].

However, the age-associated increase in these specific aAbs is not restricted to patients with ongoing infections, being also observed in apparently healthy populations. Indeed, the prevalence of anti-IFN-I aAbs neutralizing 100 ng/mL of IFN-α and/or IFN-ω increased from 1.1% in individuals under the age of 70 years old to more than 4.4% in those over the age of 70 years old, and up to 7.1% for those with aged between 80 and 85 years old [5,123,137]. The reasons why this prevalence decreases in patients aged >85 are not clear; one cause can be the fact that most of individuals died before the COVID-19 pandemic from other illnesses aggravated due to the presence of the aAbs, with only aAbs-negative ones remaining only [123]. Another plausible explanation is that “long-lived” individuals have passed a certain threshold of age selection and, therefore, their physiological parameters might differ noticeably from those of the general population. This possibility is supported by the increased percentages of naive T cells in the CD4+ T-cell subset, higher prevalence of low frequency clonotypes, and slightly higher T-cell receptor diversity observed in healthy individuals with an average age of 82 years old compared to younger subjects [181].

However, all currently reported information reported is in contrast with trends recently demonstrated in healthy individuals, in whom the number of aAbs increased with age from infancy to adolescence, when they reached a plateau [159]. This specific observation suggests that while the response to infectious agents (and may be vaccines) might contribute to the production of aAbs through molecular mimicry, this mechanism does not appear to continue to accumulate aAbs throughout life.

### 4.3. Are aAbs Detected during Infections Dangerous or Protective?

The ability of anti-IFN-I aAbs to neutralize soluble IFN-I from binding to their receptor on the surface of cells was proposed as the most straightforward mechanism through which these aAbs could promote virus replication and subsequent disease [182]. However, the significance of aAbs production and their function was not definitively established, since they may mediate diverse immunological functions depending on their specific interaction with the target cytokine. Circulating cytokine/aAbs immune complexes are probably in equilibrium with their free cytokine and free aAbs in concentrations that vary based on the levels of cytokine that need to be neutralized. In addition, their concentration, epitope specificity, avidity, isotype, and subclass may influence the capacity of these molecules to neutralize their related cytokine. It was also suggested that aAbs produced against cytokines may play a role in the physiological regulation of their biological activities via neutralizing the targets or prolonging the half-life [183].

Anti-cytokine aAbs with a moderate affinity for self-antigens provide a first line of defense against infections, probably have housekeeping functions, and contribute to the homeostasis of the immune system [4]. Furthermore, they may play a direct pathogenic role in susceptibility to infection, rather than arising as an immune response to the pathogenic micro-organism, because, in APS-1, they may be detected prior to the development of the associated infectious disease [184]. It is important to take into account that the biological significance of anti-cytokine aAbs must be evaluated in the context of disease, because they may play a role in modulating disease activity in autoimmune conditions and may also increase susceptibility to infections in certain immune-deficient patients [183]. In addition, the function of anti-cytokines aAbs during infections could theoretically be very different: they could be beneficial, harmful, or have both effects. The final case is supported by a large body of evidence indicating that aAbs induced during malarial infection are associated with disease severity and clinical outcomes, but are also capable of mediating protection against *Plasmodium* sp. [185]. It was also proposed that aAbs against inflammatory cytokines might protect against untoward inflammation [4] and be generated when such a response is of benefit for the host. On the contrary, the new aAbs, including anti-cytokine aAbs, which were recently detected in severe COVID-19, may directly cause harm, such as blood clotting, blood vessel inflammation, and tissue damage [136], and seemed to be associated with long-COVID symptoms [186], even if these data are controversial [155]. These aAbs perturb immune function and impair virological control via inhibiting immunoreceptor signaling and altering peripheral immune cell composition [128].

Finally, it is important to remember the possibility that the production of aAbs can be limited in time and can stop when the triggering stimulus fails. For instance, while anti-IFN-I aAbs remained stable in patients with AIRE deficiency and thymic malignancies, anti-IFN-I aAbs with neutralizing activity peaked soon after COVID-19 onset and declined to undetectable levels during convalescence [187]. Therefore, depending on context, these aAbs may serve beneficial “housekeeping” functions through removing surplus danger signals from circulation or, conversely, induce disease emergence.

The detection of aAbs can be useful for early diagnosis and prognosis, as observed in cardiovascular diseases and cancers, because they may be detected well in advance of clinical manifestations, enabling earlier identification of patients that may benefit from effective treatments with a targeted approach [188,189]. Therefore, aAbs found at an early stage of COVID-19 and non-COVID-19 infections seem to predict the disease severity and possible long-term effects, thus potentially facilitating more effective therapy [52,136]. Accordingly, anti-cytokine aAbs (e.g., antibodies against IFN-α, IFN-ϵ, IL-6, IL-22, GM-CSF, and TNF-α) were proposed for COVID-19 and non-COVID-19 treatment, although with different results, since some improved clinical outcomes, while others had no benefit [190]. For instance, subcutaneous IFN-β treatment of hospitalized patients did not seem to improve COVID-19 clinical outcomes [130]. Interestingly, in a phase II clinical trial, a single dose of pegylated IFN-III as an early antiviral treatment for COVID-19 did not inhibit or increase B-cell antibody responses measured in plasma, instead accelerating viral clearance [191]. Therefore, it was proposed that IFN-III could potentially be a superior choice of treatment compared to other IFN-I in SARS-CoV-2-infected patients [192].

Recently, the presence of aAbs against specific chemokines in Italian and Swiss subjects was found of help in the identification of convalescent individuals with favorable acute and long-COVID disease courses. Anti-chemokine monoclonal antibodies derived from these individuals block leukocyte migration and, thus, may be beneficial through modulation of the inflammatory response [153].

## 5. Conclusions

The extent of the COVID-19 pandemic, the large availability of biological samples enriched by clinical information, and the allocation of a large amount of research funds and dedicated staff rapidly advanced our knowledge of the balance between direct SARS-CoV-2-induced damage and inflammatory responses triggered by the virus and how this balance contributes to the broad spectrum of disease severity. The studies carried out subsequently made possible to detect a high prevalence of anti-IFN-I aAbs in both COVID-19 and the serum of patients with other non-SARS-CoV-2 viral, bacterial, and fungal infections.

aAbs can have different roles in these diseases and can act both as enemies or friends, i.e., capable of acting in a beneficial or harmful way. Establishing the levels and stability of aAbs could be useful to discriminate between the two possibilities, as it would be help us to understand the characteristics of the other components of the immune system of subjects in whom aAbs are identified. Indeed, their discovery in elderly subjects indicates that they may be linked to other defects related to immunosenescence or inflammaging.

Several questions remain to be answered regardin the significance of the presence of anti-IFNs aAbs in healthy individuals and infected patients. For instance, it must be elucidated why naturally-occurring anti-IFN-I aAbs were mainly described as binding the IFN-β subtypes and/or IFN-ω, while aAbs against IFN-β or other IFN types, such as IFN-III, appeared to be much rarer [5,87,158]. Furthermore, it must be determined whether there might be a normal physiological role for low or transient levels of anti-IFN-I aAb. Therefore, future studies on different infections and involving a larger number of patients are desirable to define the impact, long-term duration, and clinical implications of the production of such aAbs. Understanding the immunological mechanisms underlying the production of aAbs could improve the approach to some infections, focusing not only on pathogens, but also on a possible low degree of autoimmunity in these patients.

## Figures and Tables

**Table 1 viruses-15-01215-t001:** Types of IFNs.

	Other Designation	Official Gene Definition	Chromosome	Protein	Receptor
Type I IFNs					IFNAR1; IFNAR2
IFN-alpha	IFN-α1, -α2, -α4, -α5, -α6, -α7, -α8, -α10, -α13, -α14, -α16, -α17, -α21	IFNA1, IFNA2, IFNA4, IFNA5, IFNA6, IFNA7, IFNA8, IFNA10, IFNA11P, IFNA12P, IFNA13, IFNA14, IFNA16, IFNA17, IFNA20P, IFNA21, IFNA22P	9p21.3	19 kDa, 20 glycosylated, 165–166 a, 188–189 aa (human)	
IFN-beta	IFN-β	IFNB1		20 kDa, 22 kDa glycosilated, 166 aa, 187 aa (human)	

IFN-omega	IFN-ω	IFNW1		22 kDa glycosilated, 187 aa, 195 aa (human)	
IFN-epsilon	IFN-ε	IFNE		24.4 kDa, 187 aa, 208 aa (human)	
IFN-kappa	IFN-κ	IFNK		19 kDa, 182 aa 207 aa (human)	
IFN-zeta	IFN-ζ (limitin)			21.7 kDa glycosilated, 182 aa	
IFN-tau	IFN-τ	IFNT		19–24 kDa, 172 aa	
IFN-nu	IFN-v	IFNNP1		NA	
Type II IFN			12q15	17 kDa, 115–175 aa, 166 aa (human)	IFNGR1; IFNGR2
IFN-gamma	IFN-γ	IFNG			
Type III IFNs					
IFN-lamda	IFN-λ1 (IL-29)	IFNL1	19q13.2	21 kDa, 23–35 glycosilated, 200 aa (human)	IFNLR1; IL-10R2
	IFN-λ2 (IL-28A)	IFNL2		22 kDa, 24 glycosilated, 200 aa (human)	
	IFN-λ3 (IL-28B)	IFNL3		21 kDa, 24 glycosilated, 196 aa (human)	
	IFN-λ4	IFNL4		179 aa	

Greatest amount of information was taken from Negishi et al. [53] and GeneCards^®^: The Human Gene Database [54].

**Table 2 viruses-15-01215-t002:** Number and percentage of positive samples for anti-IFN-I aAbs.

COVID-19 Gravity	Numbers	Country	IFN-I aAbs Tested	Neutralizing Activity against IFNs	Total	Percentage	Publication Year	Reference
Recovered	19	Colombia	IFN-α	No	5	26.3	2021	[127]
Severe	18		IFN-α	No	3	16.7		
All	172	USA	IFN-α and IFN-ω	Yes	9	5.2	2021	[128]
All	210	The Netherlands	IFN-α and IFN-ω	No	35	17	2021	[129]
All	35		IFN-α and IFN-ω	Yes	6	17		
Severe	47	Spain	IFN-α and IFN-ω	10 ng/mL	5	10.6	2021	[130]
Critical	16		IFN-α and IFN-ω	10 ng/mL	3	18.7		
Severe	47		IFN-β	10 ng/mL	0	0		
Critical	16		IFN-β	10 ng/mL	0	0		
Convalescent	116	USA	IFN-α	No	4	3.0	2021	[131]
Convalescent	116		IFN-α and IFN-ω	10 ng/mL	2	1.5		
Critical	26	France	IFN-α and IFN-ω	Yes	8	30.7	2021	[132]
Severe	44	Italy	IFN-α, IFN-ω and IFN-β	Yes	2	4.5	2021	[133]
Critical	135			Yes	23	17		
Severe	84	France	IFN-α	No	21	25	2021	[134]
			IFN-α and IFN-ω	Yes	15	18		
Severe	623	Consortium	IFN-α and IFN-ω	10 ng/mL	22	3.53	2021	[87]
Critical	3136		IFN-α and IFN-ω	10 ng/mL	307	9.8		
Severe	522		IFN-α and IFN-ω	100 pg/mL	34	6.5		
Critical	3595		IFN-α and IFN-ω	100 pg/mL	489	13.6		
Severe	187		IFN-β	10 ng/mL	0	0		
Critical	1773		IFN-β	10 ng/mL	23	1.3		
All	8	USA	IFN-α and IFN-ω	No	1	12.8	2021	[135]
N/A	51	USA and Germany	IFN-α	No	23	45	2021	[136]
Severe	102	Several	IFN-α	10 ng/mL	6	6	2021	[137]
Critical	26		IFN-α	10 ng/mL	5	19		
Critical	275	Spain	IFN-α and IFN-ω	no	49	17.8	2021	[138]
Critical	275			10 ng/mL	26	9.5		
Severe	49	USA	IFN-α and IFN-ω	Yes	4	8.2	2021	[139]
Critical	86	Russia	IFN-α and IFN-ω	No	9	10.5	2021	[140]
Critical	47	France	IFN-α and IFN-ω	Yes	2	4.2	2022	[141]
Critical	139	France	IFN-α and IFN-ω	no	107	77	2022	[142]
			IFN-α and IFN-ω	10 ng and 100 pg/mL	11	7.9		
			IFN-β	10 ng/mL	0	0		
Deceased	11		IFN-α, IFN-ω	10 ng/mL	6	55		
Severe	70	Russia	IFN-α	no	13	18	2022	[143]
Severe	97	The Netherlands	IFN-α	Yes	7	7	2022	[144]
Fatal	38			Yes	5	13		
Severe	52	Belgium	IFN-α	?	8	15.3	2022	[145]
All	360	Italy	IFN-α	No	27	7.5	2022	[146]
			IFN-α	Yes	13	3.6		
			IFN-β	No	37	10.3		
			IFN-β	Yes	1	0.3		
Critical	237	Germany	IFN-α and IFN-ω	Yes	18	7.5	2022	[147]
Severe	235	Japan	IFN-α and IFN-ω	10 ng/mL	5	2.1	2022	[148]
Critical	170		IFN-α and IFN-ω	10 ng/mL	10	5.9		
Severe	235		IFN-α and IFN-ω	100 pg/mL	6	2.6		
Critical	170		IFN-α and IFN-ω	100 pg/mL	18	10.6		
Critical	103	Switzerland	IFN-α and IFN-ω	yes	11	10.7	2022	[107]
			IFN-β	yes	0	0		
Severe/ critical in SLE	16	France	IFN-α	10^2^ pg/mL	4	25	2022	[149]
	13		IFN-ω	10^2^ pg/mL	4	31		
	12		IFN-β	10^4^ pg/mL	2	17		
Critical	925	France	IFN-α and IFN-ω	Yes	96	10.3	2022	[124]

SLE: Systemic lupus erythematosus.

**Table 3 viruses-15-01215-t003:** Characteristics of male and female patients tested for presence of anti-IFN-I-aAbs.

	Males	Females	*p*-Value
Critical patients, number (%)	219 (52)	130 (45)	0.092
Age (years), mean (±SD)	72 ± 13	75 ± 12	0.023
Age (years) range	40–98	38–99	-
Anti-IFN-I aAbs, number (%)	28 (13)	16 (12)	1.000
Age (years), mean (±SD)	75 ± 10	78 ± 14	0.332
Deceased, number (%)	10 (36)	8 (50)	0.525
ICU	14 (50)	3 (19)	0.057
Days of hospitalization, mean (±SD)	27 ± 20	23 ± 16	0.477
Vaccinated for SARS-CoV-2, number (%)	14 (50)	10 (63)	0.534
Cardiovascular diseases	13 (46)	5 (31)	0.361
Hypertension	13 (46)	8 (50)	1.000
Dyslipidemia	8 (29)	0 (0)	0.036
Diabetes	11 (39)	2 (13)	0.089
Solid tumor	5 (18)	4 (25)	0.702
Neurologic diseases	6 (21)	6 (38)	0.303
COVID-19 complications	25 (89)	14 (88)	1.000
Acute respiratory distress syndrome	25 (89)	14 (88)	1.000

Analyses of aAbs against IFN-α2 and IFN-ω were performed using ELISA method [5,87], with few modifications. Means were compared via *t*-test, while proportions were compared via Fisher’s exact test (differences with *p* < 0.05 are considered significant and shown in bold font).

## Data Availability

Not applicable.

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
