# Peer review of "Autoantibodies to Interferons in Infectious Diseases"

_viruses, 2023, doi:10.3390/v15051215_

Round 1
Reviewer 1 Report
Dear Authors:
Report on the review article entitled “Autoantibodies to cytokines in infectious diseases.”
It is valuable to write a review article on an interesting topic related to some ambiguous autoantibodies to cytokines.
Summarizing the data from different published manuscripts and discussing the role of autoantibodies in infectious and non-infectious diseases pathogenesis is a novel narrative review article.
All the review was focusing on Interferons and other cytokines are not addressed sufficiently, so either you can change the title to “autoantibodies to Interferons” instead of cytokines or write sufficiently on some other cytokines as well as interferons.
I have some comments that I wish that it can be helpful:
- (Line 138): There is no reference for any data in Table 1.
- (Lines 185 - 194): can the author provide any evidence from literature or a speculation of the immune balance in the normal physiologic and pathogenic conditions regarding balance between interferons and their autoantibodies? This example would be very descriptive. And an imaginary graph would be helpful.
- Conclusion is not in line with the sequential narration of the context; I think that the review would be ended by briefing the story of Autoantibodies for cytokines and its role in diseases pathogenesis as in the review title.
- (Line 362 & 365): grammatical correction They instead of we. because it refers to (Busnadiego et al., 2022).
Minor spelling mistakes, just an example mentioned in the report
Reviewer 2 Report
This is a very timely and scholarly article that will be of significant interest to readers. I only have a few minor comments:
1. Is the type of the immunoglobulin anti-cytokine response almost always of the IgG subclass. The expectation would be yes =, especially where there is neutralizing activity.
2. As properly discussed, the appearance of abs increases with age. Have there been any unusual reports of cytokine autoantibodies in young people?
3. Although not mentioned in the article, have anti-cytokine receptors been found in some clinical conditions? This reviewer does appreciate that this subject could be a topic of a review in itself.
The greek "gamma" is missing on line 108 and in Table 1.
Reviewer 3 Report
In this manuscript, Quiros-Roldan et al discusses the identification of autoAbs that target cytokines in infectious diseases. The review is mainly focused on anti-IFNs Abs. It is well written and should of interest for the readers of Viruses.
Here are some concerns:
Autoantibodies targeting type I interferons have been described in patients infected with early variants of SARS-CoV-2. I was wondering if these autoAbs are still detectable in patients infected with the current variants. If studies addressing this topic exist, I think it would be interesting to discuss it in this review.
Typos:
Title: line 104: “Anti-cytokines aAbs have BEEN found…”
Line 108: “IFN-« symbol gamma is lacking, idem line 137, Table 1….
Line 456: Change “Plasmidium »
Line 467: sentence cut in half
Minor editing of English language required
Round 2
Reviewer 1 Report
Nice work and good luck.